# Analysis of NO$_2$ and O$_3$ Total Columns from DOAS Zenith-Sky Measurements in South Italy

**Paolo Pettinari** [1,2,*] , **Antonio Donateo** [3] , **Enzo Papandrea** [2] , **Daniele Bortoli** [2,4,5] , **Gianluca Pappaccogli** [3] and **Elisa Castelli** [2]

1   Dipartimento di Fisica a Astronomia, Universitá di Bologna, viale Berti Pichat 6/2, 40127 Bologna, Italy
2   National Research Council (CNR), Institute of Atmospheric Sciences and Climate (ISAC),
    via Piero Gobetti 101, 40129 Bologna, Italy
3   National Research Council (CNR), Institute of Atmospheric Sciences and Climate (ISAC),
    Str. Prov. Lecce-Monteroni, km 1.2, 73100 Lecce, Italy
4   Earth Remote Sensing Laboratory (EaRSLab), University of Evora, 7000-671 Evora, Portugal
5   Institute of Earth Sciences (ITC) and Department of Physics, University of Evora, 7000-671 Evora, Portugal
*   Correspondence: p.pettinari@isac.cnr.it

**Abstract:** The Gas Absorption Spectrometer Correlating Optical Difference—New Generation 4 (GAS-COD/NG4) is a multi-axis differential optical absorption spectroscopy (MAX-DOAS) instrument which measures diffuse solar spectra at the Environmental-Climate Observatory (ECO) of the Italian research institute CNR-ISAC, near Lecce. The high-resolution spectra measured in zenith-sky configuration were used to retrieve the NO$_2$ and O$_3$ vertical column densities (VCDs) from March 2017 to November 2019. These good-quality data, proven by the comparison with the Ozone Monitoring Instrument (OMI) and TROPOspheric Monitoring Instrument (TROPOMI) satellite measurements, were used to characterize the ECO site by exploiting the sinergy with in situ NO$_2$ and O$_3$ concentrations and meteorological data. Although stratospheric processes seem to be the main forces behind the NO$_2$ and O$_3$ VCDs seasonal trends, diurnal variabilities revealed the presence of a tropospheric signal in the NO$_2$ VCDs, which had significant lower values during Sundays. Comparison with wind data acquired at the ECO observatory, at 20 m above the ground, revealed how NO$_2$ VCDs are influenced by both tropospheric local production and transport from the nearby city of Lecce. On the other hand, no significant tropospheric signal was contained in the O$_3$ VCDs.

**Keywords:** NO$_2$; O$_3$; DOAS; Lecce

## 1. Introduction

Two of the most important trace gases for atmospheric chemistry and physics are nitrogen dioxide (NO$_2$) and ozone (O$_3$). Since they are important pollutants in urban areas, investigations of them have been carried out worldwide, focusing on both their temporal development [1,2] and their interrelations with chemical mechanisms [3]. As pollutants, NO$_2$ and O$_3$ affect human health by increasing the risk of respiratory symptoms [4,5]. Moreover, upper tropospheric O$_3$ acts as a green-house gas by absorbing long-wave terrestrial radiation [6,7]. Nitrogen oxides (NO$_x$), defined as group made of nitric oxide (NO) and NO$_2$, are released into the atmosphere from both natural and anthropogenic sources. Major NO$_x$ sources include fossil fuel combustion, which is particularly important in urban environments [8]; biomass burning; lightning; and oxidation of ammonia [9,10]. NO$_2$ is mainly transformed by oxidation to nitric acid [11]. Although combustion processes directly emit only small quantities of NO$_2$, they release nitrogen monoxide (NO), which rapidly (few minutes) forms NO$_2$ when reacting with ozone molecules. Nitrogen dioxide participates, among others, in catalytic cycles leading to tropospheric ozone (O$_3$) formation. It also acts as a catalyst for the stratospheric O$_3$ destruction [12] and contributes to the formation of secondary aerosols [13].

Differential optical absorption spectroscopy (DOAS) is a technique which allows to derive information on atmospheric trace gases, exploiting spectra acquired in the visible and UV spectral ranges [14]. DOAS experimental setups have been widely used for the observation of atmospheric trace gases in the past decade both from satellite and ground-based instruments [15–20]. From the ground, the zenith-sky DOAS and multi-axis-DOAS (MAX-DOAS) configuration measurements allowed us to retrieve columnar and vertically resolved information of atmospheric trace gases, respectively, and to validate satellite measurements [21,22]. Since satellite measurements provide valuable information for the investigation of the spatial distribution of air pollutants, they have been used to investigate the transport and dynamics of emissions from both anthropogenic and natural sources. However, significant underestimation of the tropospheric $NO_2$ vertical columns was reported for the Ozone Monitoring Instrument (OMI) and TROPOspheric Monitoring Instrument (TROPOMI) satellite observations when compared to ground-based measurements [23–25]. To solve that issue, recent studies have shown how satellite data can be significantly improved by exploiting further simulated and measured information [26–28].

However, since the temporal resolution of satellite measurements is typically low, it is useful to compare and integrate them with ground-based DOAS data for the interpretation of the spatial and temporal variation of $NO_2$ and $O_3$.

The purpose of this paper is to provide a detailed characterization of the ECO measurement site in terms of $NO_2$ and $O_3$ vertical column densities (VCDs), in order to highlight the importance of ground-based DOAS data, in synergy with other in situ measurements, for the air quality monitoring. This site could also become important for satellite validation purposes because, at the moment, it hosts the only DOAS instrument that gathered a multi-year data series in South Italy.

In the paper, we present the VCDs of atmospheric $NO_2$ and $O_3$ at an urban background station located in South Italy for the years 2017, 2018 and 2019, using the custom-built DOAS instrument Gas Absorption Spectrometer Correlating Optical Difference—New Generation 4 (GASCOD/NG4). Details about the experimental site, instrument, satellite data used and analysis methods are provided in Section 2. Results and discussions are described in Sections 3 and 4, respectively, and conclusions are drawn in Section 5.

## 2. Materials and Methods

### 2.1. Site Description

The routine observational measurements were carried out at the Environmental Climate Observatory (ECO) of Institute of Atmospheric Sciences and Climate (ISAC)—Italian National Research Council (CNR). The ECO observatory (40.34°N 18.12°E; 36 m a.s.l.) is located about 4 km (W–SW) from the urban area (about 95,000 inhabitants), about 10 km from the South Adriatic sea and can be classified as an "urban background" site [29,30]; see Figure 1. The site is located about 30 and 80 km from the two most important industrial centers of the Apulia Region (Brindisi and Taranto, respectively). The ECO observatory is a regional station of the Global Atmospheric Watch (GAW) network [31]. The spectrometer was located on a shelter on the roof of the ISAC-CNR building, 12 m above the street, inside the university campus. At the ECO observatory, detailed ancillary meteorological data (wind characteristics, temperature, relative humidity and pressure) and in situ $NO_2$ and $O_3$ concentrations are measured by an automatic weather station (Vaisala WXT520), located 20 m above the ground, and a gas analyzer described in [31], respectively. For the interpretation of the results, wind data and in situ $NO_2$ and $O_3$ concentrations were analyzed together with the retrieved $NO_2$ and $O_3$ VCDs.

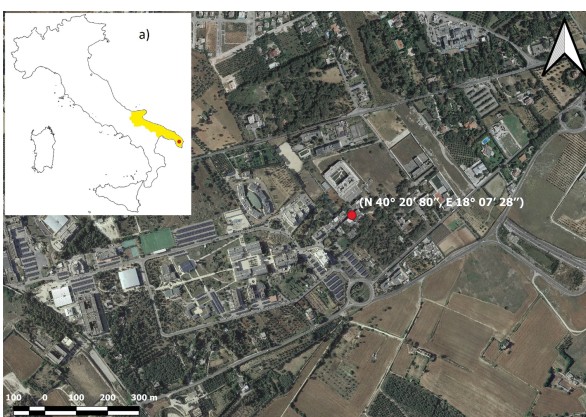

**Figure 1.** Map of the measurement area indicating the observation site (red circle) at the ECO observatory. In the inset "a)" is the location of the Apulia region (in yellow).

### 2.2. Instruments

#### 2.2.1. MAX-DOAS GASCOD/NG4

The ground-based instrument used in this work is part of the GASCOD family [32,33]. This kind of instrument was initially developed at CNR–ISAC in the 90s and improved over the years in collaboration with the University of Evora (Portugal) [34]. Several instruments of this family are located in permanent observation stations: the Ottavio Vittori facility at Mt. Cimone (44.11°N, 10.42°E—Italy) [35,36], the "Mario Zucchelli" station (74.70°S, 164.12°E—Antarctica) [34], the atmospheric observatory in Bologna (44.52°N, 11.42°E—Italy) and the Evora Atmospheric Science Observatory (EVASO) in Evora (38.56°N, 7.90°W—Portugal) [33]. In the scope of the I-AMICA project (http://www.i-amica.it, accessed on 2 November 2022), the meteo-climatic station in Lecce (Italy) was equipped with the GASCOD/NG4 (New Generation Rev.4) in 2016. This instrument uses the same monochromator as all the other systems installed in the above mentioned stations, with the dispersive element of 1200 grooves/mm and mean spectral resolution of 0.5 nm in the 300–800 nm spectral range. A detailed description of the mechanical and electronic components, together with the adopted optical layout and explanations of the measurement principles adopted in the most recent versions of these instruments, are available in [34,37].

#### 2.2.2. OMI

The Dutch–Finnish instrument OMI was successfully launched onboard the US National Aeronautics and Space Administration (NASA) Earth Observing System (EOS) Aura spacecraft on the 15 July 2004. The Aura sun-synchronous polar orbit has an inclination of 98.2° and an altitude of 705 km and a local afternoon equator crossing time (ascending node) at 13:45 and completes 14 orbits per day [38]. The instrument is a nadir-looking, push broom ultraviolet/visible solar backscatter grating spectrometer. The light entering the telescope is depolarized using a scrambler and then split into two channels: the ultraviolet channel (UV, from 270 to 380 nm) and the visible channel (VIS, from 350 to 500 nm) [39]. The instrument telescope has a viewing angle of 114° so that the two-dimensional detector measures with a wide swath of 2600 km on the Earth's surface in the across track direction of the satellite. In the normal global operation mode, the OMI ground pixel at nadir is $13 \times 24$ km [38]. Among several targets of the OMI mission, we can list the monitoring of the ozone ($O_3$) layer and the observation of trace gases such as nitrogen dioxide ($NO_2$), sulfur dioxide ($SO_2$) and formaldehyde (HCHO). For this reason, OMI contributed to research regarding the anthropogenic and natural emissions on local-to-global scales and the transport of pollution [39–41].

### 2.2.3. TROPOMI

The TROPOMI instrument is aboard the European Space Agency (ESA) low-Earth-orbit polar satellite Sentinel-5 Precursor (S-5P), launched on the 13 October 2017. The S-5P orbit is a near-polar frozen sun-synchronous orbit with an inclination of approximately 98.7°, a mean local solar time at ascending node (LTAN) of 13:30 h, a repeat cycle of 17 days and a nominal height of 824 km. TROPOMI has inherited the peculiar features from both the OMI [38] and the Scanning Imaging Absorption Spectrometer for Atmospheric Cartography (SCIAMACHY) [42]. TROPOMI is a nadir–viewing hyperspectral system and has four separate spectrometers measuring the ultraviolet–visible (UV–VIS, 270–500 nm), the near–infrared (NIR, 675–775 nm) and the short–wavelength infrared (SWIR, 2305–2385 nm) radiation; the NIR and SWIR bands are new compared to its predecessor OMI [43]. The instrument images, on a two-dimensional detector, a strip of the Earth with dimensions of approximately 2600 km across track and 7 km along track and has a very high spatial resolution: a ground pixel at nadir of 7 km × 3.5 km before 6 August 2019 and 5.5 km × 3.5 km afterwards. This characteristic enables to better capture the high spatial variability of pollutants which occurs in the lower troposphere than OMI, especially over urban sites [41]. TROPOMI measures key atmospheric constituents, including $O_3$, $NO_2$, $SO_2$, carbon monoxide (CO), methane ($CH_4$), HCHO and aerosol properties [43]. A more detailed description of the TROPOMI experiment can be found in [43].

### 2.3. Analysis Method

#### 2.3.1. DOAS Methodology

The retrieval of atmospheric trace gases from UV–Vis radiation measurements is widely performed through the use of the DOAS technique that exploits the structured absorption of many trace gases in the measured spectral ranges. While the DOAS method was originally developed for ground-based measurements [14,44], it now has a wide range of applications for measurements from different platforms, including aircraft, satellite and cars. The DOAS method is based on the differential approach: the logarithm of the ratio between two spectral radiances measured at different times and/or for different geometries is linked to the gaseous optical thickness along the light path. The differential approach does not require any radiometric calibration when both the measured spectra are acquired by the same instrument. Then, the broad and narrow spectral structures are separated. Gaseous features (represented by the absorption cross-sections) contribute to the narrow structures. The cross-sections are fitted to the logarithm of the ratio of the two spectra, and the fitting coefficients represent the differential slant column densities (SCDs). The SCD, that is, the gas concentration integrated along the light path, depends on the solar zenith angle (SZA) and the geometry of observations, and it is influenced by the presence of aerosols and clouds. The SCDs of a certain gas can thus be converted into VCDs through the use of the air mass factors (AMFs). The AMFs can be estimated using radiative transfer models (RTM) and depend on the wavelength, the SZA, the trace gases' input profiles (mainly the one we are interested in), the viewing geometry and the presence of clouds and aerosol. A detailed description of the whole procedure is given in the following subsections.

#### 2.3.2. DOAS Data Analysis: QDOAS Elaboration

Zenith-sky spectra, acquired by the GASCOD/NG4, were analyzed by the QDOAS software (http://uv-vis.aeronomie.be/software/QDOAS, last accessed on 16 September 2022) in order to retrieve the $O_3$, $NO_2$ and tetraoxygen ($O_4$) SCDs. The automatic acquisition system of GASCOD/NG4 acquires the spectra in different spectral windows and continuously during the day. The system saves the data into binary files containing, for each spectrum, the information about the measured spectral range, day and time of acquisition and SZA. Those files are pre-processed to spectrally calibrate the measurements and to make them compliant with the QDOAS input file format. The analysis reported in this work was performed in the spectral interval 470–510 nm, exploiting the spectra acquired in the spectral window centered at 486 nm (see Figure 2). In this spectral region,

absorption features of $NO_2$, $O_3$ and $O_4$ are present. $O_4$ SCDs are important because they can be used for the detection of aerosol and clouds that can alter the $NO_2$ and $O_3$ SCDs (see Section 2.3.3).

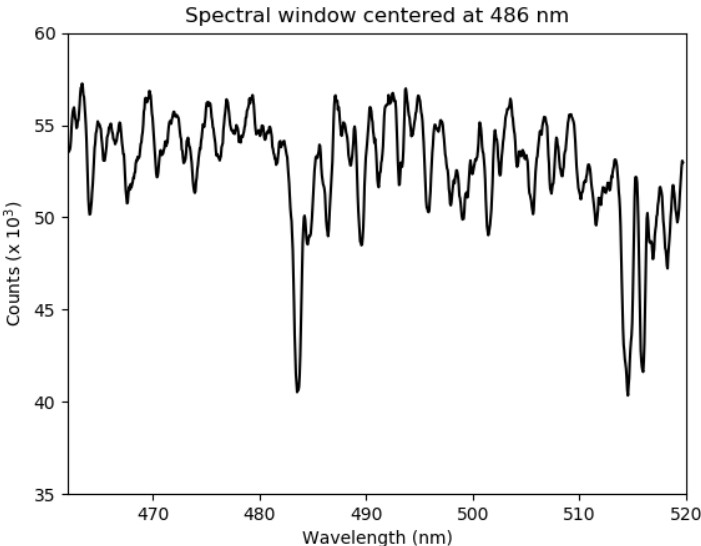

**Figure 2.** Zenith spectrum acquired at 11:27 UTC of 27 November 2018 by GASCOD/NG4 in the spectral window centered at 486 nm in arbitrary units. It was not radiometrically calibrated, since DOAS analysis does not require it.

The QDOAS settings used for the analysis follow as much as possible the Fiducial Reference Measurements for DOAS (FRM4DOAS) community requirements.

At first, all the spectra were analyzed with respect to a fixed reference spectrum. However, we realized that, due to several maintenance interventions voted to improve the instrument performance (high SNR), the dispersion parameters applied to the spectral images reaching the CCD sensor slightly changed, resulting in the modification of the GASCOD/NG4 spectral resolution (see Figure 3). The spectral resolution was evaluated by QDOAS during the automatic calibration procedure. The differential method at the basis of the DOAS technique requires the spectral resolutions of reference and analyzed spectra to be as similar as possible.

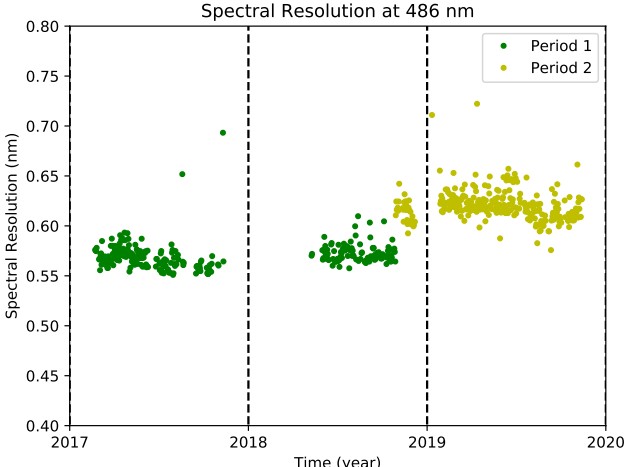

**Figure 3.** The instrument's spectral resolution in the spectral window centered at 486 nm and for all the measurements periods. Each color identifies a period characterized by a time-constant spectral resolution.

Hence, we divided the analysis into two different periods (see the colors in Figure 3), each of them with a fixed reference spectrum, as reported in Table 1. QDOAS gives also the possibility of performing the analysis with respect to daily reference spectra, automatically measured every day around noon time. However, we decided to use only two fixed reference spectra for two main reasons:

- The use of daily reference spectra would have introduced daily biases in the retrieved VCDs, due to the uncertainty in the knowledge of the true contributions of the reference spectra. According to our methodology, we expected biases only between the two analysis periods.
- The two fixed reference spectra, acquired in summer at noon, are affected by a minimum gas absorption due to the almost vertical position of the sun.

**Table 1.** Reference spectra in the two different periods used for QDOAS analysis.

| Period | beginning—28/10/2018 | 29/10/2018—end |
|---|---|---|
| Day ref. | 06/07/2017 | 11/07/2019 |
| SZA ref. | 17.62° | 18.13° |
| NO$_2$ Ref. (molec/cm$^2$) | $4.3 \times 10^{15}$ | $6.1 \times 10^{15}$ |
| O$_3$ Ref. (molec/cm$^2$) | $7.1 \times 10^{18}$ | $1.3 \times 10^{19}$ |

All the other analysis settings are reported in Table 2. A constant offset between the analyzed and reference spectra and an order-3 polynomial were fitted simultaneously with NO$_2$, O$_3$, O$_4$ and H$_2$O absorption cross-sections. The ring effect is considered as an additional cross-section (details in [45]). Since NO$_2$ and O$_3$ cross-sections depend on temperature, the absorption signature for each of the gases was fitted by two cross-sections at different temperatures. Moreover, NO$_2$ and O$_3$ theoretical cross-sections were corrected through the convolution with I$_0$ correction, as suggested in [46].

**Table 2.** Main QDOAS settings used for NO$_2$ and O$_3$ SCDs analysis.

| Wavelength Range | 470–510 nm |
|---|---|
| **Polynomial** | Order 3 |
| **Offset** | Constant |
| **Cross sections** | |
| NO$_2$(220 K) | from [47]. I$_0$ correction ($10^{17}$) applied |
| NO$_2$(294 K) | from [47]. Orthogonalized to NO$_2$ (220 K) with I$_0$ correction ($10^{17}$) |
| O$_3$(223 K) | from [48]. I$_0$ correction ($10^{20}$) applied |
| O$_3$(293 K) | from [48]. Orthogonalized to O$_3$ (223 K) with I$_0$ correction ($10^{20}$) |
| O$_4$(293 K) | from [49] |
| H$_2$O(298 K) | from [49] |
| Ring | Generated according to [45], using the solar atlas in [50] |

An example of the differential optical paths, due to the NO$_2$, O$_3$ and O$_4$ absorption, fitted by QDOAS, is reported in Figure 4. In this case, the NO$_2$, O$_3$ and O$_4$ spectral signatures are well defined compared to the fit residuals because the path crossed by the radiation in the analyzed spectrum (measured at a high solar zenith angle) is much longer than the one relative to the reference spectrum (measured with a low solar zenith angle; see Table 1).

Before the calculation of the VCDs, the SCDs retrieved by the QDOAS were filtered. First, SCDs were filtered out according to the QDOAS flag that certified whether the fit was successful or not and according to the $\chi^2$ of the fit. Then, a second filter was applied in order to exclude data heavily contaminated by clouds. The remaining SCDs were then processed to compute the VCDs.

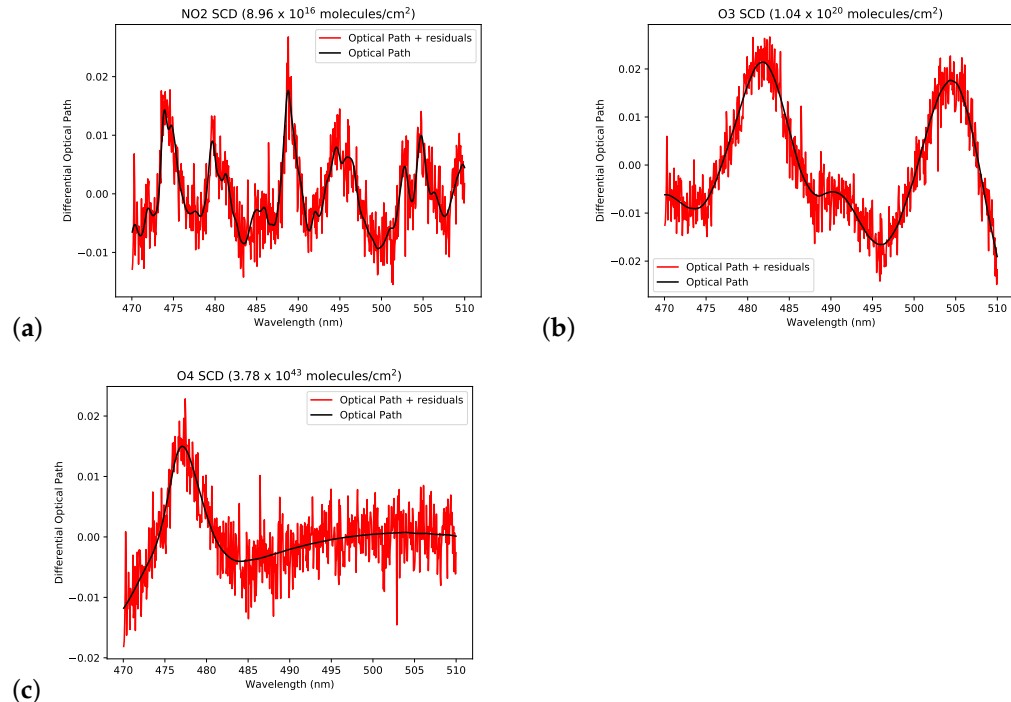

**Figure 4.** Examples of differential optical paths due to $NO_2$ (**a**), $O_3$ (**b**) and $O_4$ (**c**) absorption, fitted by QDOAS for a spectrum acquired in 6 July 2017 at 19:11 UTC when the SZA was 89.3°. SCDs values are reported in the plots' titles.

### 2.3.3. Clouds and Aerosol Data Filtering

Since SCDs are path-integrated quantities, variations in the light path due to scattering by particles produce biased SCDs and thus VCDs. However, as demonstrated by [51], $O_4$ SCDs can be used to infer information on particles' optical depths and vertical distribution. The $O_4$ concentration in the atmosphere is well known (its absorption is proportional to the square of the partial pressure of molecular oxygen) and constant. $O_4$ SCDs variations depend on the light path, and its behavior as a function of SZA is thus known. Variations in this behavior from the expected one are proxies of particle presence.

In this study, we used $O_4$ SCDs to filter clouds/aerosol-contaminated measurements. The $O_4$ SCDs' behavior with respect to SZA can be modeled using a RTM, e.g., SCIA-TRAN [52]. However, in several cases, as reported by [53], the simulated $O_4$ SCDs can differ (as a bias and not in the behavior) from the measured ones. For this reason, to filter the data, we decided to use only the measured $O_4$ SCDs. The data were used to build histograms, binning the $O_4$ SCDs in 2°-wide SZA bins. For each SZA bin, data falling outside 90% of the maximum frequency were filtered out. We should mention that, using this filter procedure, particle-contaminated spectra may still be present. Indeed, this method aims at filtering only spectra heavily contaminated by particles to remove strong oscillations from the final dataset.

The $O_3$ and $NO_2$ SCDs identified as particle-contaminated were removed from the subsequent analysis. In the end, a total of 16.6% of the measurements were filtered out.

### 2.3.4. Reference Contributions

Since the absorber amounts in the two reference spectra used for the QDOAS analysis cannot be considered negligible as they would be with spectra measured outside the atmosphere, the SCDs estimated by QDOAS had to be corrected by adding the reference contributions that can be estimated using the Langley plot method [54]. According to it, the SCDs, retrieved for a certain period, were plotted against the AMFs. In the hypothetical case that the true absorber VCDs remain constant in time, the intercept of the line, which

fits the SCDs, represents the reference contribution, and the slope represents the constant VCD value related to the observed SCDs. However, the assumption of a constant vertical content is not easy to satisfy, especially close to an urban area, such as at our measurement site. For this reason, we adopted the same approach described in [55]. We assumed that, in a Langley plot, the lowest SCDs over a certain period refer to the same minimum vertical content. This means that the minimum SCDs can be linearly fitted in order to estimate the intercept, as reported in Figure 5. The $NO_2$ and $O_3$ estimated SCDs in the two selected reference spectra are reported in Table 1.

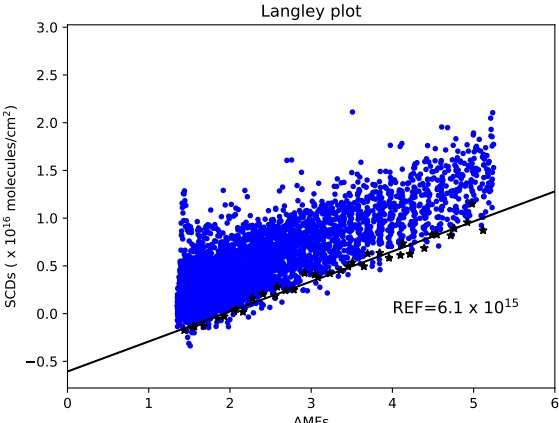

**Figure 5.** Langley plot used to estimate the contribution of the reference spectrum acquired at 11 July 2019 to the $NO_2$ SCDs. Blue dots are all the $NO_2$ SCDs relative to March 2019. Black stars are the low SCDs, fitted by the black line, which were used to estimate the intercept.

### 2.3.5. Retrieved VCDs

VCDs were then calculated by dividing the corrected SCDs (after adding the reference contribution) by the corresponding AMFs simulated by the SCIATRAN RTM. Standard profiles (included in the SCIATRAN code) representative for the ECO observatory latitudes and accounting for seasonal variations of temperature, pressure and trace gases concentrations were used.

The results of our analysis, after the filtering process, provided a total of 81,310 $NO_2$ and $O_3$ VCDs retrieved in 592 days between March 2017 and November 2019. On average, 137 VCDs were retrieved for each day.

### 2.3.6. Systematic Errors Affecting the VCDs' Diurnal Variability

The estimation of the diurnal variability of trace gases vertical columns derived from zenith-sky DOAS measurements is a hard task because the systematic errors, mainly in the estimated reference contributions and in the simulated AMFs, have a not-negligible impact. The errors in the $NO_2$ and $O_3$ reference contributions were estimated as $2 \times 10^{15}$ and $2 \times 10^{18}$ molecules/cm$^2$, respectively. These values were computed as the spread coming from reference estimates performed considering different periods. This error has a higher impact on the retrieved VCDs relative to low AMFs (around noon) than the ones with high AMFs (sunrise or sunset). This artifact contributes to creating non-real diurnal variability in the VCDs. For this reason, the assessment of the reference contributions, discussed in Section 2.3.4, is a sensitive part of the analysis.

The errors in the simulated AMFs lead to a similar problem. Actually, the input parameter which most affects the simulated AMFs is the vertical profile of the target gas used in the RTM. This effect can be seen in Figure 6a for $NO_2$ and 6b for $O_3$, where the percentage differences between AMFs simulated with modified input profiles and standard profiles (the input for simulating the used AMFs) are plotted against the SZA. The changes in the input profiles were obtained by increasing (multiplying to 10 or 100) or decreasing (dividing by 10 or 100) the tropospheric contents below 3 km. The plots show that the

AMFs did not change significantly when $NO_2$ and $O_3$ tropospheric contents were decreased. This means that the tropospheric content in the standard profiles is low, and the relative AMFs are mainly representative of the stratosphere. On the other hand, an increase in the tropospheric content led to important differences in the AMFs. In particular, the AMFs changed differently for low and high SZAs, contributing to creating non-real diurnal variation in the VCDs. Since this effect can heavily affect the diurnal behavior of $NO_2$ and $O_3$ VCDs, we tried to make some considerations for the diurnal variabilities without focusing on the absolute values.

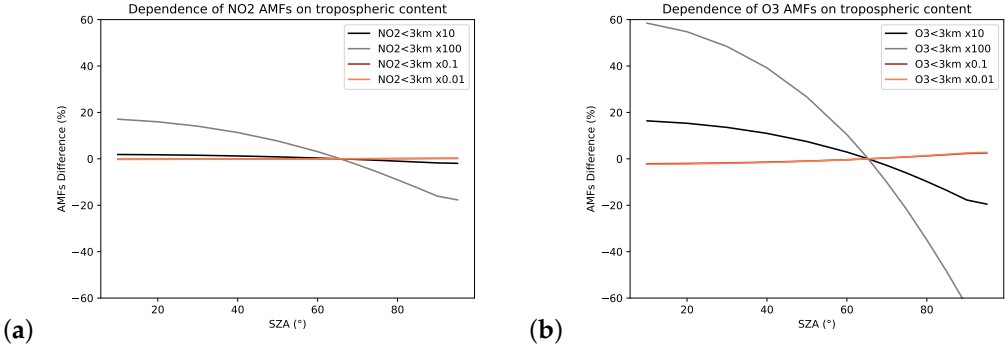

(**a**)  (**b**)

**Figure 6.** Percentage differences between AMFs computed using perturbed input profiles and standard profiles, for $NO_2$ (**a**) and $O_3$ (**b**). The AMFs simulated with standard profiles were the ones used to convert the SCDs into VCDs.

### 2.3.7. Selection of Coincident Satellite Data

The OMI $NO_2$ Standard Products (OMNO2) V4.0 [56] were downloaded from the NASA Goddard Earth Sciences Data and Information Services Center (https://search.earthdata.nasa.gov/search?q=OMNO2_003, last accessed 16 September 2022). They contain $NO_2$ SCDs (total amount along the average optical path from the sun into the atmosphere and then toward the satellite), the total $NO_2$ VCDs, the stratospheric and tropospheric VCDs, AMFs, scattering weights for calculation of AMFs and other ancillary data. The $NO_2$ column content included in the data was derived using the DOAS technique on the UV-VIS hyperspectral earthshine radiance measurements in the range 400–470 nm, where $NO_2$ has a strong, structured absorption feature. More details can be found in [56]. In the OMI product files, the effective cloud fraction (CF) for each ground pixel is reported. We used these values to discriminate between clear and cloudy measurements, indicating with $qa = 1 - CF$, the quantifier which ranges from 0 (cloudy) to 1 (clear).

For the considered periods, TROPOMI offline (OFFL) products) http://doi.org/10.5270/S5P-s4ljg54, last accessed on 16 September 2022), available in the NetCDF format, were considered. They were downloaded using the Copernicus Data Hub from https://s5phub.copernicus.eu/dhus (last accessed on 16 September 2022), with the help of a bash script which exploits the capability of the "open data protocol" interface for accessing the Earth Observation (EO) data stored on the archive. These products contain the $NO_2$ total column content, derived using the DOAS method applied to the UV–VIS backscattered solar radiation measurements in the 405–465 nm wavelength range [57]. In the TROPOMI product files, each ground pixel has a "quality assurance value" (hereafter reported as qa) associated with it, a continuous variable, which can assume continuous values from 0 (no output) to 1 (everything is fine). According to [57], the users should use a qa threshold of 0.75 to be sure that clouds, scenes covered by snow/ice, errors and problematic retrievals are removed. However, it is also possible to consider a qa threshold of 0.50, if the purpose is to select good quality retrievals with clouds or snow/ice in view.

## 3. Results

### 3.1. Diurnal Variability

Although, as mentioned in Section 2.3.6, the NO$_2$ and O$_3$ diurnal trends are affected by important systematic errors, Figure 7a clearly shows that significant day-to-day differences in the NO$_2$ VCDs exist. In particular, it is evident how the NO$_2$ VCDs were higher and less constant on 14 July 2018 (Saturday) with respect to 22 July 2018 (Sunday), when the traffic was generally lower. This result highlights the clear presence of the tropospheric contribution to the retrieved NO$_2$ VCDs. On the other hand, the O$_3$ VCDs retrieved in the same two days were not affected by any significant difference (Figure 7b), suggesting that the O$_3$ VCDs are less sensitive to the tropospheric variability. These first considerations are further demonstrated in Sections 3.2 and 3.3.

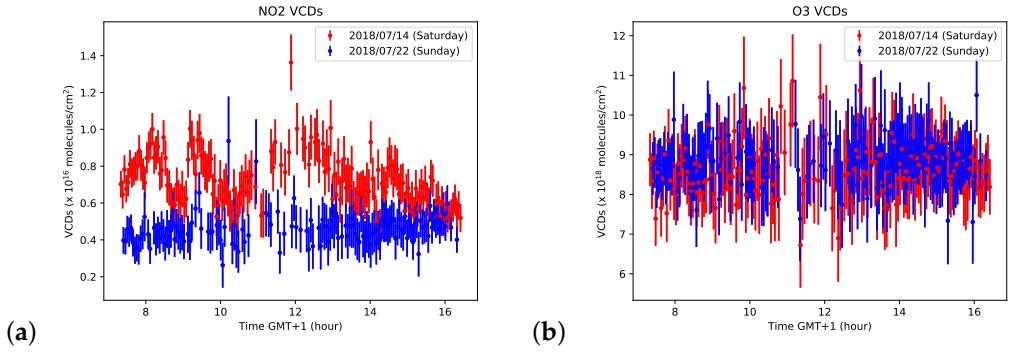

**(a)**                                                                 **(b)**

**Figure 7.** NO$_2$ (**a**) and O$_3$ (**b**) VCDs during the days 2018/07/14 (red) and 2018/07/22 (blue). The error bars represent the VCDs random errors derived from the SCDs fitted by QDOAS.

In Figure 8a, we can see that, for most days, the NO$_2$ VCDs in the afternoon are higher than the ones in the morning; the median and mean differences are $6.2 \times 10^{14}$ and $5.1 \times 10^{14}$ molecules/cm$^2$, respectively. These differences were computed as the mean NO$_2$ VCDs in the SZA range between 60° and 90° in the afternoon minus the same quantity in the morning. The same fixed SZA range, used to compute the mean VCDs in the morning and afternoon, avoids the results being affected by the systematic effects, due to the AMFs, as discussed in Section 2.3.6. This diurnal increase is in agreement with the stratospheric chemical processes that involve NO$_2$, where, during daytime, N$_2$O$_5$ is photolyzed into NO$_2$ and NO$_3$. A slight and linear diurnal increase in NO$_2$ VCDs is indeed found in not-polluted areas, such as over Table Mountain, California, from direct solar spectra [54], and in Zugspitze, Germany, from solar FTIR measurements [58]. Thus, this analysis suggests that the diurnal variability of the NO$_2$ VCDs over the ECO observatory is the result of both contributions related to the stratospheric chemistry and to the tropospheric pollution due to the anthropogenic activity. The same analysis, performed for the O$_3$ VCDs (Figure 8b), shows that, differently from the NO$_2$, the O$_3$ columns decrease during most days. The median and mean differences between afternoon and morning are $-5 \times 10^{17}$ and $-7.5 \times 10^{17}$ molecules/cm$^2$, respectively.

### 3.2. VCDs vs. Day of the Week

Since the results shown in Figure 7a suggest the presence of an important tropospheric variability in the NO$_2$ VCDs, data were analyzed while taking into account the day of the week, in order to see if significant differences due to the different anthropogenic activity (mainly traffic) between weekends and working days exist. For this purpose, we computed the daily mean VCDs in the time range between 8:00 and 16:00 local time. These average data were then used for the computation of a 7-day moving mean, in order to filter out the day-to-day variability. For each day, we calculated the difference between the daily mean VCD and the 7-day running average, derived around the considered day. This operation was performed for each day for which data are present for all 7 days around it ($\pm$3). These

differences were used to compute the mean anomaly for all the seven days of the week. The $NO_2$ and $O_3$ VCDs mean anomalies are plotted against the days of the week in Figure 9a.

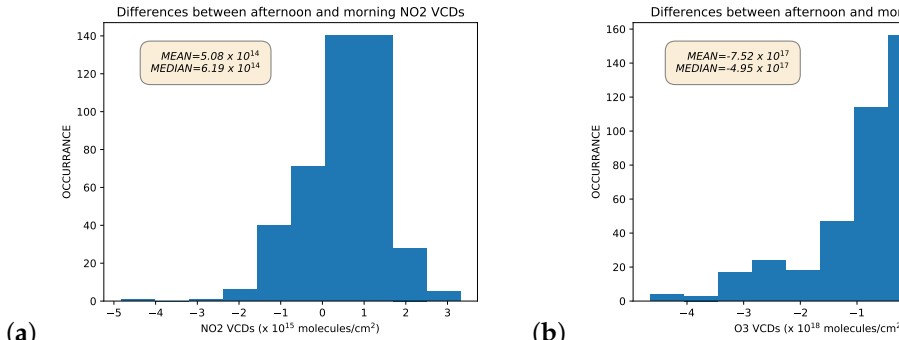

**Figure 8.** Histogram of the differences computed as $NO_2$ (**a**) and $O_3$ (**b**) VCDs in the afternoon minus the VCDs in the morning. For each day, VCDs representative of morning and afternoon were computed as the mean VCDs in the SZA range between $60°$ and $90°$.

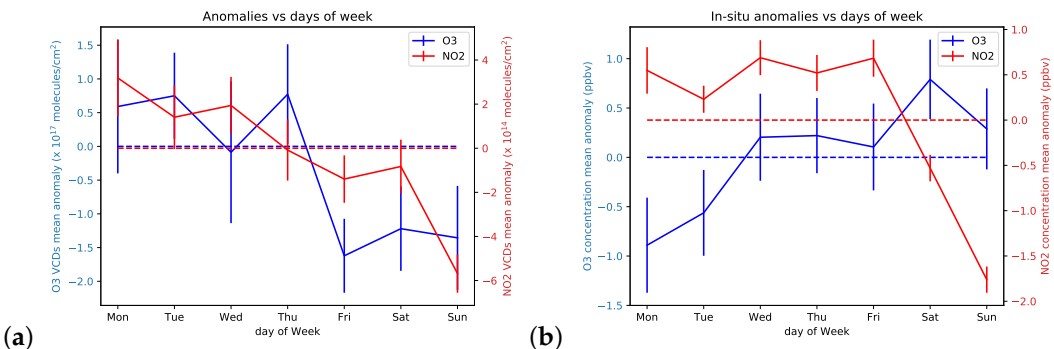

**Figure 9.** Mean anomalies of $NO_2$ and $O_3$ VCDs (**a**) and in situ concentrations (**b**) in each day of the week. The error bars are standard deviations of the means. The dashed lines represent the conditions with a null anomaly.

$O_3$ VCDs anomalies are higher during working days, decreasing during the weekends. However, these anomalies are not highly significant because they have similar magnitudes compared to the large error bars. On the other hand, $NO_2$ VCDs had a significant negative anomaly of about $-6 \times 10^{14}$ molecules/cm$^2$ for Sunday, confirming the presence of anthropogenic signal.

In order to better understand and to make the obtained results more significant, the same procedure was applied to the $NO_2$ and $O_3$ concentrations measured in situ by the gas analyzer installed at ECO observatory. The $NO_2$ results in Figure 9b are in agreement with the weekly variability observed for the $NO_2$ VCDs, indicating important $NO_x/NO_2$ production due to traffic, as also reported for other cities [59,60]. On the other hand, the $O_3$ in situ measurements reveal a slight increasing trend over the week, in contrast with the $O_3$ VCDs weekly trend. However, the high error bars, compared to the $O_3$ in situ anomalies, highlight again the low significance of the results.

*3.3. $NO_2$ VCDs vs. Wind at 20 m*

In this subsection, we show the results regarding the correlation between the retrieved $NO_2$ VCDs and the wind speed and direction measured at an altitude of 20 m at the ECO observatory.

Figure 10a shows the $NO_2$ VCDs anomalies, computed as described in Section 3.2, with respect to the diurnal mean wind velocities estimated in the same time range. Even though data are scattered and the correlation is low, the negative slope of the fitting line ($-0.18$ molecules s m$^{-1}$ cm$^{-2}$) confirms that the retrieved $NO_2$ VCDs contain a tropospheric

signal, a consequence of the anthropogenic emissions, and states that one of the main causes of polluted days at ECO observatory is the local production. On the other hand, no plot with O$_3$ VCDs is reported in this section because no correlation was found with the wind speed at 20 m altitude. This result confirms again the lack of an O$_3$ tropospheric signal.

Figure 10b shows the NO$_2$ VCDs mean anomalies with respect to the wind direction. It highlights that, together with NO$_2$ local production, a transport contribution from the city of Lecce exists as well. Indeed, the peak around 50° (about NE) corresponds to the direction where the city is located with respect to the ECO observatory.

As in Section 3.2, the same analysis was also performed with NO$_2$ in situ data. Figure 11a shows that the NO$_2$ in situ concentrations anomalies decreased with the wind speed, with a slope of $-0.51$ ppbv s m$^{-1}$. This result is in agreement with the one obtained for the NO$_2$ VCDs, even though the slopes are quite different. We must take into account, indeed, that these two quantities are hardly comparable.

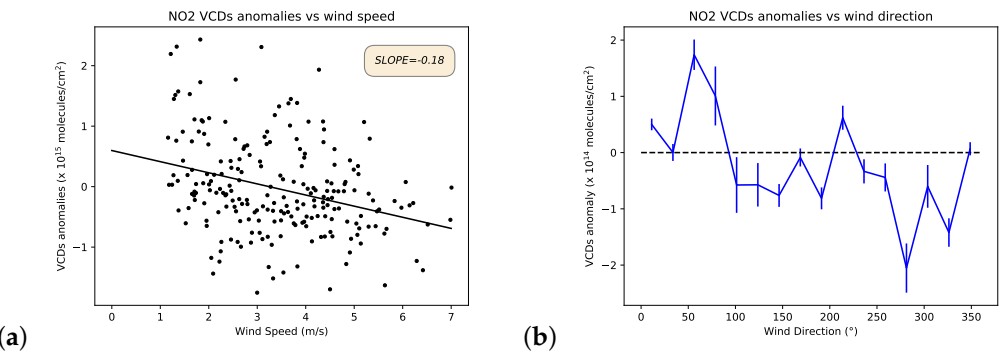

**(a)**  **(b)**

**Figure 10.** NO$_2$ VCDs anomalies with respect to the wind speed (**a**) and wind direction (**b**) measured at 20 m. In figure (**b**), the wind directions are indicated clockwise with 0° representing the north. Lecce is located in the northeast direction, where the peak is found.

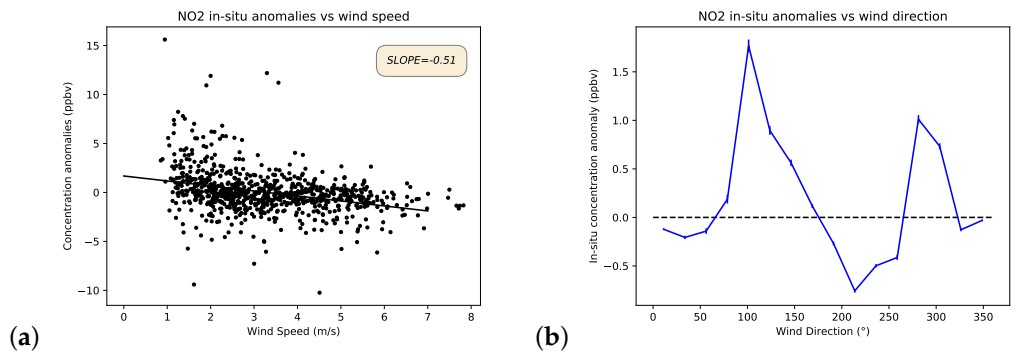

**(a)**  **(b)**

**Figure 11.** NO$_2$ in situ concentrations anomalies with respect to the wind speed (**a**) and wind direction (**b**) measured at 20 m.

The analysis of NO$_2$ in situ concentration anomalies in relation to the wind direction (Figure 11b) shows completely different results compared to the ones obtained for the NO$_2$ VCDs in Figure 10b. The peak found for the VCDs and located around 50°, in the direction of Lecce, is not present in the in situ concentrations. However, two new peaks, around 100° and 290°, appeared. This discrepancy probably occurs because since NO$_2$ VCDs are sensitive to the transport in the whole boundary layer, they are affected by a long-range transport and the signal coming from the city of Lecce is detectable. On the other hand, since in situ concentrations are representative of a smaller spatial radius, the two peaks in Figure 11b are probably consequence of the presence of streets around the ECO observatory.

### 3.4. Seasonal Variability

Figure 12 shows the NO$_2$ (a) and O$_3$ (b) monthly averages of the VCDs retrieved in the years 2017, 2018 and 2019. Monthly averages were computed using only data acquired

within the SZAs range between 60° and 80° to mitigate the systematic effects derived from AMFs simulations, as shown in Figure 6.

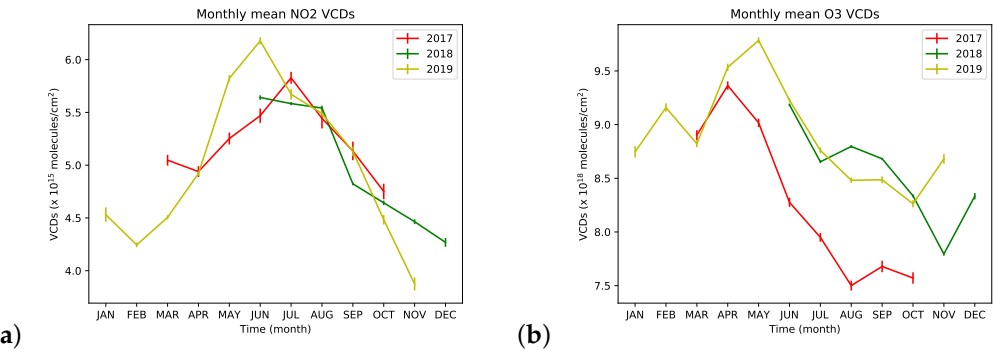

**Figure 12.** Monthly variability of NO$_2$ VCDs (**a**) and O$_3$ VCDs (**b**) in the three analyzed years. The error bars represent the standard deviations of the monthly mean values.

The NO$_2$ monthly mean VCDs ranged between $3.5 \times 10^{15}$ and $7 \times 10^{15}$ molecules/cm$^2$. All the three years were characterized by similar behavior: the NO$_2$ VCDs increased during spring, reaching the maximum value around June and July, and then started to decrease at the end of summer. Although the dataset does not contain much information for the winter periods, we can clearly notice that the decrease continued until November (2018, 2019) or December (2018). During March, the NO$_2$ VCDs started to increase (2019). This behavior is typical of the stratospheric NO$_2$ seasonal cycle in the middle latitudes, as reported in [61], which is related to the number of sunlit hours.

Further, the O$_3$ VCDs seasonal variability is mainly driven by processes occurring in the stratosphere, where most of the O$_3$ is present. Its monthly mean VCDs ranged between $7.5 \times 10^{18}$ and $1.0 \times 10^{19}$ molecules/cm$^2$. The values started to increase during autumn, reaching the maximum around spring, and then rapidly decrease during summer.

### 3.5. Comparison with Satellite Data

The overpass times of OMI and TROPOMI are similar, being 13:45 and 13:30 local time, respectively. For each satellite overpass, all the pixels located within a radius of 20 km around the position of the station were selected. The 20 km radius was chosen in order to produce a robust analysis, considering the available satellite pixels/day in a homogeneous scene (the pixels are only over land and at the ground level). This criterion led to selecting for TROPOMI, for each day, a maximum number of 51 pixels before the 6 August 2019, as the ground pixel size was 7 km × 3.5 km, and 64 afterwards, as the along-track pixels' size was reduced from 7 to 5.5 km. Due to the higher size of the OMI ground pixel (13 km × 24 km), no more than six OMI pixels could be selected for each day. For this comparison exercise, all satellite data which had a quality flag lower than 0.5 were filtered out. More conservative quality filtering, with a threshold of 0.75, could also have been implemented. However, we have verified that the comparison results were not significantly affected.

Figure 13 reports the mean NO$_2$ VCDs, computed in a 3-h time bin around the satellite overpass time, measured by GASCOD/NG4 (blue dots), and the corresponding OMI (green line in the panel (a)) and TROPOMI (red line in the panel (b)) NO$_2$ VCDs averaged within the 20 km radius. We can observe overall good agreement, both in the absolute values and in the ability to capture the seasonality. The same averaged NO$_2$ VCDs are used in the scatterplots of Figure 14.

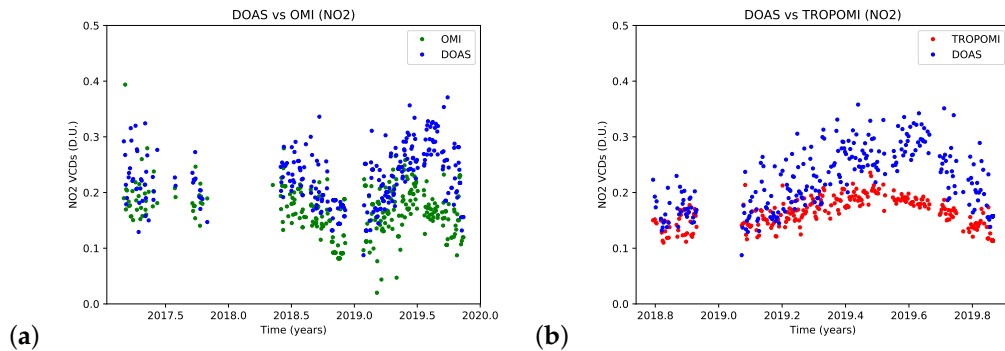

**Figure 13.** NO$_2$ VCDs measured by GASCOD/NG4 (blue dots in both panels), OMI (green line in the panel (**a**)) and TROPOMI (red line in the panel (**b**)) over the ECO observatory. DOAS measurements are time averages in a 3-h time bin around satellite overpass, and satellite data are the averages of all the coincidences within a 20 km radius.

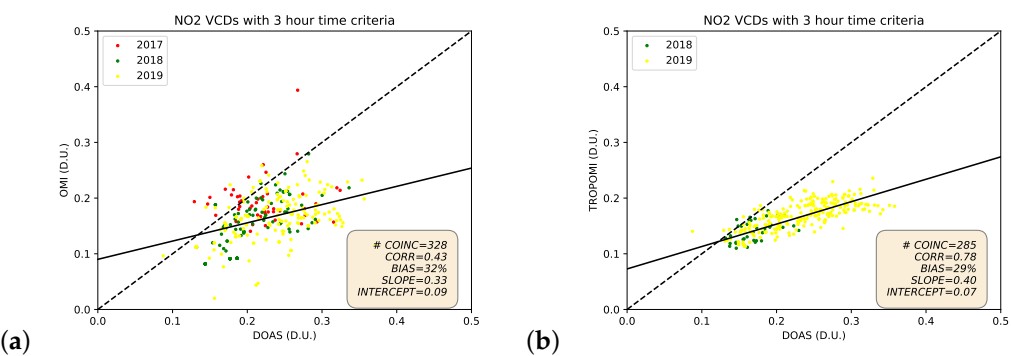

**Figure 14.** Scatterplot of NO$_2$ VCDs measured by GASCOD/NG4 versus OMI (**a**) and TROPOMI (**b**). DOAS data were averaged in a 3-h time bin around satellite overpass and are compared with the mean satellite VCDs in the same time bin and within a radius of 20 km from the ECO observatory.

The NO$_2$ VCDs retrieved from GASCOD/NG4 are, on average, 32% higher than OMI and 29% higher than TROPOMI. This positive bias is also evident in the timeseries in Figure 13. Moreover, the two scatterplots show that important differences exist between OMI, which provides more scattered NO$_2$ columns, and TROPOMI. In particular, DOAS results are in a better agreement with TROPOMI, with a correlation coefficient of 0.78 compared to 0.43 found with OMI. The estimated linear regression slopes confirm that both satellite measurements underestimate the high values of NO$_2$ VCDs detected by GASCOD/NG4. A similar result was found in [24], which compares the NO$_2$ VCDs measured by TROPOMI and a ground-based Pandora instrument in Boulder (Colorado). They showed that very good agreement exists during low-pollution conditions and that TROPOMI underestimates the NO$_2$ VCDs by about 30% in the presence of high NO$_2$ concentrations. The authors of [24] performed a comparison among different measurement sites, and they found that the TROPOMI underestimation is more pronounced in the most polluted cities. In [25], both OMI and TROPOMI NO$_2$ products are compared with the ones measured by a ground-based MAX-DOAS instrument in the Jing–Jin–Ji region (China). This study revealed that both OMI and TROPOMI underestimate the NO$_2$ columns by about 30% to 50%, in this highly polluted region. In accordance with the cited literature, our comparison showed good agreement during low polluted conditions.

The same comparison was performed for O$_3$ VCDs, and the results are presented in Figures 15 and 16. General good agreement in magnitude and seasonality occurred also in the O$_3$ VCDs, with correlation coefficients of 0.65 and 0.67 with respect to OMI and TROPOMI, respectively. The percentage biases found in both the O$_3$ comparisons (3 and 4%) are lower than the ones found for NO$_2$ (about 30%), showing better agreement. Other

validation results revealed a lower mean bias between ground-based and satellite $O_3$ VCDs, in the order of 1–2% [62,63]. However, these values, obtained for latitude bands, came with high standard deviations (4–5%) that justify the higher biases in local regions.

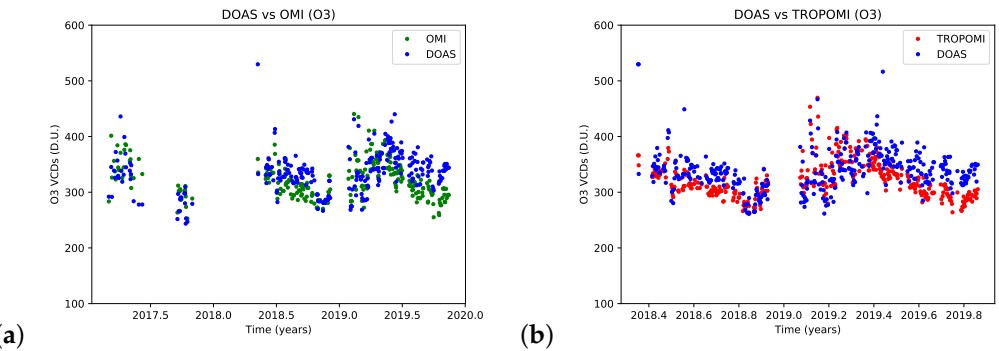

**Figure 15.** $O_3$ VCDs measured by GASCOD/NG4 (blue dots in both panels), OMI (green line in the panel (**a**)) and TROPOMI (red line in the panel (**b**)) over the ECO observatory. DOAS measurements are time averages in a 3-h time bin around satellite overpass, abd satellite data are the averages of all the coincidences within a 20 km radius.

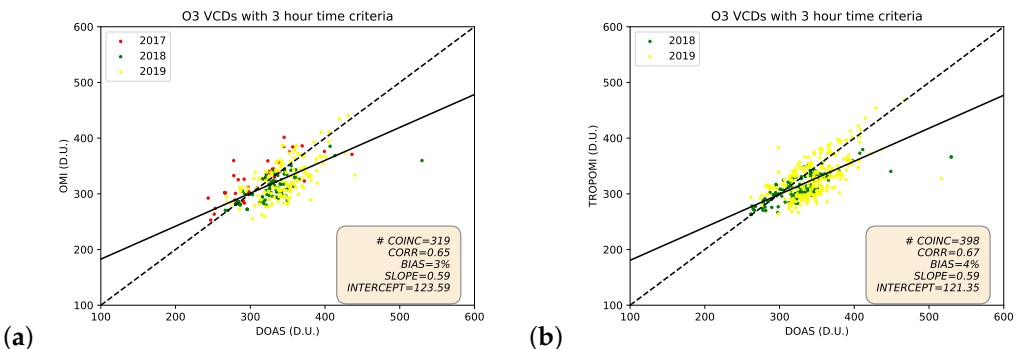

**Figure 16.** Scatterplot of $O_3$ VCDs measured by GASCOD/NG4 versus OMI (**a**) and TROPOMI (**b**). DOAS data were averaged within the 3-h time bin and are compared with the mean satellite VCDs in the same time bin and within a radius of 20 km from the ECO observatory.

## 4. Discussions

Although the results have shown that the $NO_2$ VCDs are influenced by both stratospheric and tropospheric contents, it is a hard task to quantitatively divide the two contributions with DOAS zenith-sky measurements. However, some qualitative considerations can be made. The analysis in Section 3.2 has shown that the tropospheric $NO_2$ has a minor contribution on Sundays. Further, Figure 7a highlighted that, considering the differences between two days (Saturday 14 July 2018 and Sunday 22 July 2018), the tropospheric $NO_2$ significantly affects the VCDs during polluted days. Moreover, since the $NO_2$ VCDs during Sunday (Figure 7a) do not present highly variable peaks, as occurs in Saturday, it is likely that they represent the stratospoheric VCDs, which in July, have values in the order of $4 \times 10^{15}$ molecules/cm$^2$.

The DOAS measurements also reveal that the $O_3$ VCDs in 2017 are systematically lower than in all the other years; the differences are in the order of $1 \times 10^{18}$ molecules/cm$^2$ (see Figure 12b). This discrepancy cannot be fully explained by the systematic errors in the reference spectra contributions because such a difference is also present between data acquired in 2017 and 2018, which were analyzed with respect to the same reference spectrum. This result was partially confirmed by OMI data. Indeed, we have verified that OMI $O_3$ VCDs in 2017 are also lower than the ones in 2018 during the months from June to October. As an example, after computing the averages $O_3$ VCDs for August 2017 and 2018 from OMI data, we saw that the mean value in 2018 is higher than that of 2017 by about

16 D.U., which corresponds to about $4.3 \times 10^{17}$ molecules/cm$^2$. This is more or less half of the difference that we found in the DOAS O$_3$ VCDs (see Figure 12b).

We have also verified that the results of the comparison between ground-based DOAS VCDs and satellite measurements are robust because they do not significantly depend on the spatial radius chosen to compute the averages around the ECO observatory, as summarized in Table 3. Indeed, similar and very stable results were found for the comparisons with TROPOMI. On the other hand, the comparisons with OMI revealed less stable statistical parameters, partially justified by the important decrease in coincidences with the chosen radius.

We must mention that the systematic errors described in Section 2.3.6 could be responsible for some of the positive biases found in the comparison results. Indeed, taking into account Figure 6 and considering that OMI and TROPOMI passes over Lecce at around noon, the systematic effect due to the AMFs would lead to an overestimation of the retrieved VCDs, mainly when the SZAs around noon are low (summer) and when high NO$_2$ and O$_3$ tropospheric concentrations are present. This theory is confirmed in Table 4, where all the statistical parameters for the four different seasons are reported. Indeed, it is evident that the NO$_2$ and O$_3$ biases have seasonal behavior, increasing during summer and decreasing in winter. However, due to the low number of coincidences, mainly during winter, these statistical parameters are less stable and robust than the ones reported in Table 3.

The results have highlighted that the NO$_2$ tropospheric contribution significantly affects the NO$_2$ VCDs, and O$_3$ VCDs contain less strong tropospheric signals. This explains why the O$_3$ comparison with respect to satellite data showed better agreement compared to the NO$_2$ one. Indeed, satellite measurements are less sensitive to the lower part of the atmosphere.

**Table 3.** Statistical parameters of the comparisons between ground-based DOAS and satellite VCDs for different spatial criteria for the selection of satellite data.

| | NO$_2$ | | | | | | O$_3$ | | | | | |
| | OMI | | | TROPOMI | | | OMI | | | TROPOMI | | |
| **RADIUS (km)** | **20** | **10** | **5** | **20** | **10** | **5** | **20** | **10** | **5** | **20** | **10** | **5** |
| **COINCIDENCES** | 328 | 99 | 28 | 285 | 270 | 260 | 319 | 74 | 23 | 398 | 394 | 372 |
| **CORRELATION** | 0.43 | 0.51 | 0.45 | 0.78 | 0.78 | 0.76 | 0.65 | 0.50 | 0.49 | 0.67 | 0.68 | 0.66 |
| **BIAS (%)** | 32 | 28 | 24 | 29 | 28 | 27 | 3 | 4 | 2 | 4 | 4 | 4 |
| **SLOPE** | 0.33 | 0.39 | 0.47 | 0.40 | 0.39 | 0.39 | 0.59 | 0.43 | 0.55 | 0.59 | 0.62 | 0.62 |
| **INTERCEPT (D.U.)** | 0.09 | 0.08 | 0.07 | 0.07 | 0.08 | 0.08 | 124 | 179 | 143 | 121 | 111 | 111 |

**Table 4.** Statistical parameters of the comparisons between ground-based DOAS and satellite VCDs with a spatial radius of 20 km and for different seasons.

| | NO$_2$ | | | | | | | | O$_3$ | | | | | | | |
| | OMI | | | | TROPOMI | | | | OMI | | | | TROPOMI | | | |
| **SEASONS** | **WIN** | **SPR** | **SUM** | **AUT** | **WIN** | **SPR** | **SUM** | **AUT** | **WIN** | **SPR** | **SUM** | **AUT** | **WIN** | **SPR** | **SUM** | **AUT** |
| **COINCIDENCES** | 27 | 96 | 90 | 115 | 31 | 80 | 69 | 105 | 26 | 88 | 91 | 114 | 35 | 91 | 146 | 126 |
| **CORRELATION** | 0.67 | 0.25 | 0.08 | 0.49 | 0.62 | 0.70 | −0.03 | 0.74 | 0.93 | 0.62 | 0.54 | 0.44 | 0.94 | 0.65 | 0.59 | 0.60 |
| **BIAS (%)** | 17 | 25 | 34 | 38 | 14 | 25 | 37 | 30 | −3 | −1 | 7 | 5 | −4 | 0 | 7 | 8 |
| **SLOPE** | 0.53 | 0.17 | 0.06 | 0.46 | 0.31 | 0.31 | −0.01 | 0.37 | 0.91 | 0.45 | 0.40 | 0.21 | 0.91 | 0.39 | 0.38 | 0.34 |
| **INTERCEPT (D.U.)** | 0.06 | 0.14 | 0.17 | 0.05 | 0.09 | 0.10 | 0.19 | 0.07 | 41 | 197 | 181 | 227 | 44 | 214 | 189 | 183 |

## 5. Conclusions

NO$_2$ and O$_3$ VCDs were retrieved from zenith–sky spectra acquired in the visible spectral range from March 2017 to November 2019 by GASCOD/NG4, a ground-based spectrometer able to measure the solar diffuse radiation over the ECO observatory, located near the city of Lecce in Apulia region (Italy). For the VCDs derivation, the DOAS technique was applied, and the SCDs, relative to the measured spectra, were fitted using QDOAS.

The results show that the retrieved $NO_2$ VCDs are affected by both the stratospheric and tropospheric conditions, due to the anthropogenic activity. Indeed, the $NO_2$ VCDs present important day-to-day variability, typical of the tropospheric $NO_2$, and a systematic diurnal increase, in agreement with results obtained in non-polluted regions [54,58], as a consequence of stratospheric processes. A more detailed analysis has confirmed the presence of tropospheric signal in the $NO_2$ VCDs. Sundays, when the traffic is generally lower around the ECO observatory, are characterized by systematically lower $NO_2$ VCDs.

We also found out that the wind, measured at the height of 20 m, has an impact on the retrieved $NO_2$ VCDs. In particular, although the $NO_2$ VCDs decrease with the wind speed, providing information about the presence of local production, the analysis with respect to the wind direction stated that $NO_2$ transport exists as well, having a peak in correspondence with the NE direction, where the city of Lecce is located.

Most of these considerations are valid for the measured $NO_2$ in situ concentrations as well. However, the analysis of in situ concentrations related to the wind direction highlighted that the $NO_2$ transport occurs from different directions compared to that obtained from the $NO_2$ VCDs. This happens because in situ concentrations are probably more affected by local pollutant sources (for example, streets), which are located close to the ECO observatory, and are not sensitive to the pollution transported at higher altitudes from Lecce.

All the same analyses were performed for the retrieved $O_3$ VCDs as well, showing the presence of no tropospheric signal.

We also found that both $NO_2$ and $O_3$ VCDs are affected by seasonal variabilities. $NO_2$ monthly mean VCDs ranged between $3.5 \times 10^{15}$ and $7 \times 10^{15}$ molecules/cm$^2$; lower values were observed during winter, and the peak was in June or July. $O_3$ monthly mean VCDs ranged between $7.5 \times 10^{18}$ and $1.0 \times 10^{19}$ molecules/cm$^2$. The $O_3$ peak was reached during spring, around April or May; then it started to decrease, reaching the lowest values during autumn. These seasonal trends in the total columns are mainly driven by stratospheric processes which are influenced by different seasonal insolations.

The comparison between $NO_2$ and $O_3$ VCDs measured by GASCOD/NG4 and those retrieved from satellite measurements (OMI and TROPOMI) revealed good agreement with the results found in the literature. $O_3$ VCDs are affected by a bias (computed as GASCOD/NG4 minus OMI/TROPOMI) of 3–4%. Higher biases, of about 30%, were found with the $NO_2$ VCDs. Important underestimations occurred during highly polluted conditions. The correlations computed with respect to OMI and TROPOMI suggest that generally better agreement was present with respect to TROPOMI. This is more evident in the $NO_2$ results, where the correlation increases from 0.43 with OMI to 0.78 with TROPOMI, due also to the higher spatial resolution of TROPOMI.

In conclusion, the potential of the GASCOD/NG4 at the ECO observatory, part of the GAW network in South Italy, has been assessed with this level of detail for the first time through the synergy between DOAS, in situ and satellite $NO_2$ and $O_3$ data. In the future, measurements acquired at low elevation angles could be exploited to quantitatively divide the tropospheric and stratospheric columns, allowing us to better study the link between pollutant trace gases and the urban anthropogenic activity. This is a further step that we intend to take.

**Author Contributions:** Conceptualization, P.P., E.C. and A.D.; formal analysis, P.P., E.P. and E.C.; investigation, P.P.; data curation, D.B., A.D. and G.P.; writing—original draft preparation, A.D., P.P., E.C. and E.P.; writing—review and editing, P.P., E.C., E.P., A.D., G.P. and D.B.; visualization, P.P. and E.C. All authors have read and agreed to the published version of the manuscript.

**Funding:** This research was funded by I-AMICA Project (Infrastructure of High Technology for Environmental and Climate Monitoring PONa3_00363), a project funded under the Italian National Operational Program "Research and Competitiveness" 2007–2013.

**Institutional Review Board Statement:** Not applicable.

**Informed Consent Statement:** Not applicable.

**Data Availability Statement:** GASCOD/NG4 data, in situ $NO_2$ and $O_3$ concentrations and wind data acquired at ECO observatory are available upon request from the authors. OMI standard products V4.0 and TROPOMI offline products V1.0.0 (before 21 March 2019) and V1.1.0 (after 21 March 2019) can be downloaded from the websites https://search.earthdata.nasa.gov/search?q=OMNO2_003 and https://s5phub.copernicus.eu/dhus (last accessed on 20 October 2022), respectively.

**Acknowledgments:** The authors acknowledge the DOAS UV-VIS team at BIRA-IASB led by M. Van Roozendael for QDOAS. The authors also acknowledge SCIATRAN developers. The authors acknowledge Giorgio Giovanelli for his important contribution to the development of the CNR-ISAC custom-built GASCOD/NG4 instrument. The authors also acknowledge Bianca Maria Dinelli for her review contribution, which was important for the final form of the paper.

**Conflicts of Interest:** The authors declare no conflict of interest.

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
