# Peer review of "Analysis of NO2 and O3 Total Columns from DOAS Zenith-Sky Measurements in South Italy"

_remotesensing, doi:10.3390/rs14215541_

Round 1
Reviewer 1 Report
The authors present four years (2016 to 2019) of DOAS measurement data for NO2 and O3, taken at the Environmental-Climate Observatory (ECO) near Lecce, Italy, a Global Atmosphere Watch (GAW) site. The paper describes in great detail the instrument and data analysis procedure, before proceeding to some science results from sub-daily to multi-annual time scales. Specifically addressed are the different diurnal cycle on week-days and Sundays (NO2), the overall higher NO2 concentrations in the afternoon as compared to mornings, or the pronounced annual cycle of both NO2 and O3. Also presented is an analysis of the NO2 data in terms of wind direction and speed measured 20m above ground, which demonstrates a clear impact of the nearby city of Lecce. No such impact is detectable for O3. Arguments and hypothesis are given as to the relative importance of stratospheric versus (local) tropospheric contributions of NO2 and O3, respectively, to the observed signal. Comparison with satellite data from OMI and TROPOMI is invoked to put the DOAS data into perspective and examine its overall quality.
The paper is well written, fits the scope of Remote Sensing, and its contents are certainly of relevance to readers interested in quantitative assessment of the chemical composition of the stratosphere and troposphere, including air pollution questions. Given the prominent role of GAW for monitoring Earth's atmosphere, papers as the one presented here are generally highly welcomed and strongly needed by the science community.
However, the quality of the results (science) section currently lags behind the data and methods section. The individual findings, from sub-daily to annual time scales, are only marginally linked. Arguments on the relative importance of stratospheric NO2 and O3 as compared to (local) tropospheric contributions are often not too robust, taking rather the form of hypothesis than real arguments. This should be improved - and in my opinion can be done with reasonable effort, thereby also making the paper much stronger on the science side. I therefore recommend major revisions and encourage the authors to bring the science part of their paper in yet better shape, to further elaborate on some key points as detailed below.
Major concerns:
---------------------
Both my major concerns have to do with the vertical distribution of NO2 and O3.
A) Looking at Figure 6 shows the key role played by the concentration of NO2 and O3 in the lower most 3km of the atmosphere. This provokes three questions. First, what are the actual concentrations in the presented results and how do they compare to the concentrations examined in Figure 6? Second, how robustly confined are these (below 3km) concentration behind the presented results? How (im-)plausible is it that these concentrations were 10 times bigger or smaller and how would this affect results? Third, looking at Figure 6a, could it be that the measurements are not sensitive enough in the sense that low NO2 concentrations do not
have any measurable effect in the DOAS data? Can the authors elaborate on the first question throughout the paper and at least comment on the other two questions?
B) The separation of the tropospheric (<3km) and stratospheric contribution to NO2 and O3 are of scientific interest, e.g. regarding air quality. In several places in the manuscript, this point is addressed. Could these arguments be augmented by information on, for example, the average relative contribution of the stratospheric and tropospheric absorption for both, O3 and NO2? More information on the VCD and how it is obtained might help. Also, the authors may bring into play the role played by week-days (Sunday!) or wind direction. See also point A and some of the minor points below.
Minor points:
---------------------
l. 28: "nitrogen oxide (NO), which rapidly forms NO2". Where possible, it may be useful for a reader to replace / augment words like 'rapidly' with some rough time scale - minutes, hours, days etc.
l. 30: You point out the fact that satellite data tend to underestimate tropospheric concentrations, in particular of NO2. One might wonder then why you use OMI and TROPOMI data all the same. Is the underestimation systematic enough across sites for satellite data to be useful in the present context?
l. 46: It may help to clearly state the overall goal of the paper explicitly. I take that you want to provide a detailed description of the DOAS, illustrate / quantify the data quality, and give an idea of what scientific potential is in the data. This also given that the sites is a GAW site, thus having additional data available on top of the DOAS data and profiting, as a site, from a good overall characterization of site properties.
l. 87: At least in the data availability statement, please provide details on the data satellite data products you used, e.g. what version of OMI retrievals, web site etc.
l. 142: "the trace gas input profiles" What do these profiles look like? Given their crucial role for the entire paper, it may be useful to provide the reader with some more information, especially also the partitioning between troposphere (<3km) and stratosphere. Also, do you assume the same profile independent of hour of the day / day of the year etc.?
Figure 3: So the spectral resolution decreased from 2016 to 2019? Please comment.
l. 215: A total of 16.6% of measurements are filtered out as they are considered contaminated by clouds and / or aerosols. Given this rather small percentage one wonders what precise criterion you apply to decide on 'contaminated yes / no'. Intuitively, I have the impression that you must
have quite a few cloudy data points that do not qualify as 'contaminated by clouds'. Please comment.
l. 263: You argue that the sensitivity of the AMF to the vertical profile of the target gas is so strong for O3 (Figure 6b) that you renounce at examining O3 diurnal profiles. Two questions come to mind. Are you nevertheless confident that the quantitative retrievals of O3 are ok and, if so, why? Regarding diurnal cycles, in particular of O3, could one use / interpret Figure 6b in terms of "the diurnal cycle is at least (or at most)..."? After all, Figure 6b suggests a systematic dependence.
l. 263: Looking at Figure 6 again, Figure 6b prevents you from looking at the diurnal cycle for fear of having systematic biases. How confident are you that the non-sensitivity demonstrated in Figure 6a does not imply missing out a NO2 signal? That you are not sensitive enough to (low?) tropospheric NO2 concentrations? Please comment.
l. 267: "...the central part of the day, when the SZA is lower than 60.". I am a bit confused by the SZA being lower than 60, especially also when I read the caption of Figure 9, where you say SZA is between 80 and 85. To me, the SZA is the angle at solar noon between vertical lines from the center
of the Earth through the observer (first line) and the position of the sun (second line). But maybe I'm wrong here. In any case, a clear statement of what you mean with SZA would be helpful for a reader. Also, given that you look at diurnal cycles in the context of human activity (traffic etc.), it may be helpful to sometimes add information on the correspondence between SZA and time (hour of the day, GMT or local time or GMT+1; how about daylight saving time?)
l. 301: How do these VCDs compare to the values used in Figure 6? And can something be said on what fraction of the VCD is below 3km / in the stratosphere? This also with regard to your discussion here and there of whether the measured signal is rather due to the stratosphere or the troposphere (<3km).
l. 308: What do you mean by an "average diurnal trend"?
Figures 7 / 8 / 9 / 11: Comparing the four figures, a number of questions come to my mind, as detailed in the following.
Figure 8 shows a clearly different daily cycle for one Sunday as compared to one Saturday. Would the impression prevail if the average over all available Sundays and Saturdays, respectively, were taken? Would the impression prevail if the average were taken over daily anomalies? (instead of the daily cycle of the data as such, first subtract the mean value of the specific day) If Sundays generally have a more constant daily cycle (as compared e.g. to Saturdays), could Sundays be used to get a better idea of stratospheric versus tropospheric contributions?
What would Figure 7 look like if you were not to take the average over all days but, instead, were to average daily anomalies? (first subtract the daily mean value, thereby putting weight on deviations from the daily mean, while reducing the effect of days that have a very high daily mean and thus may dominate the average) What would Figure 7 look like if you were to average only over Sundays? (The thought is inspired by Figure 8) Punch line is that I am wondering whether the week-day dependence of NO2 concentrations (if robust) could not be further carved out / corroborated and then be used to say something on tropospheric versus stratospheric
contributions.
What would Figure 9 look like for Sundays only? If the underlying physical reason is indeed in the stratosphere, as the authors suggest, Sundays alone should look similarly as 'all days'. Also, how does Figure 9 (higher concentrations in the afternoon than in the morning) reflect in Figure 7? This is not so obvious to me.
Figure 9: what does a SZA range from 80 to 85 mean? See my general remark on SZA earlier on. And, to link Figure 9 with Figures 7 and 8, what (approximate) time in GMT + 1 corresponds to these SZAs?
Figure 11 / Section 3.3. may be moved before the analysis of the seasonal variability (section 3.2). To me, it seems closely connected to the material presented in section 3.1. (diurnal cycle). But up to the authors.
Figure 10: How about any systematic seasonal biases in the VCD? Is this, in contrast to the diurnal cycle of O3, no worry here?
Figure 12: Maybe label in the figure (or mention again in the caption) that Lecce is at 50 degrees? Also, how are degrees counted - counter-clock wise? What is at about 200 degrees? What at about 320 degrees and
120 degrees? Could these comparatively pristine wind directions be used to separate 'polluted boundary layer air' (lower 3km of troposphere with emissions from e.g. Lecce) from mainly stratospheric effects on the VCD?
Could you sort your data according to wind direction and examine seasonality, week-day means, daily cycles etc.?
Figure 13 and 14: could it be that biases between DOAS and satellite are larger when there is more tropospheric influence in the DOAS? Biases (Figure 14) seem larger for larger DOAS values.
Figure 15: I find it difficult to judge by eye that the OMI data in 2017 (red) is lower than in 2018 (green), as is stated in the text. Please comment.
Figure 17: DOAS and satellite are in better agreement here, for O3, than for NO2 (earlier figures). This may be in line with O3 (in contrast to NO2) being dominated by stratospheric contributions, which are more easily captured by the satellite. Put differently: Do you see with the DOAS primarily the stratosphere for O3, but primarily the troposphere for NO2 (at least for high values)? Please comment.
Conclusion section: A reader may appreciate if you point out here or earlier what of the presented findings is really new and how it compares with literature. Given that the site is a GAW site, readers would probably appreciate to know whether your study is the first one to asses (on this level of detail) O3 and NO2 at this site.
Reviewer 2 Report
The paper presents new interesting results and could be published after minor corrections, see please the review file.

Reviewer 3 Report
This manuscript explores the comparison between NO2 and O3 total columns based on different satellite retrievals (OMI and TROPOMI) and MAX-DOAS measurements, as well as the seasonal and diurnal trends of NO2 and O3 within Leece. The topic is of scientific impacts, but several modifications and details must be included before publication:
(1) Section 1 Lines 37-44: It is correct that significant underestimation of tropospheric NO2 vertical columns have been reported based on OMI or TROPOMI retrievals, however the authors should also take into account and highlight the recent improvements of OMI/TROPOMI or other satellite products for retrieving NO2 columns. Some good references are as follows:
(a) Resolving spatial gradients of NO2 in vertical NO2 columns in a better manner by incorporating a-priori profile inputs, combining with meteorological variables etc., so that more realistic AMFs are obtained.
Ref: https://www.mdpi.com/2072-4292/10/11/1789
(b) Better OMI NO2 standard product by improving surface and cloud treatments, and the enhancement of AMFs, thus providing NO2 columns with better data quality
Ref: https://amt.copernicus.org/articles/14/455/2021/amt-14-455-2021.html
(c) Use of different satellite input products with higher spatial resolution, and incorporate more realistic cloud treatment, so that better NO2 VCDs are obtained in updated TROPOMI product
Ref: https://amt.copernicus.org/articles/14/7297/2021/
(2) Section 1 (Introduction): The authors need to explain how MAX-DOAS measurements were previously adopted to show NO2 vertical distribution etc./ assessment accuracy of satellite-derived NO2 columns. Some references are:
(b) https://link.springer.com/article/10.1007/s00376-021-0370-1
(3) Section 1: The authors should explicitly mention the motivation and scientific advancement of this study. Currently, it is not quite clear.
(4) Lines 68-69: Perhaps other meteorological variables should be considered as well? Please kindly check.
(5) Lines 93-99: The authors may include a few studies that OMI is properly used or adopted for practical retrieval, similar for Section 2.2.3 (TROPOMI). Further, the key technical and practical differences in terms of usage of OMI and TROPOMI products should be discussed in this manuscript as well.
(6) Table 2: It should be "orthogonalized"? Also, what do you mean by references [42] and [43] as labelled? Are the methods directly adopted here?
(7) Section 2.3.6 (Lines 242-246): Again, please refer to the following references about bias in OMI-VCD retrieval when compared with MAX-DOAS measurements, and include some possible comparison / discussions in this sub-section:
(a) https://aaqr.org/articles/aaqr-21-12-ssea-0398
(b) https://edepot.wur.nl/470628
(8) Lines 258-267: Have you incorporated modeled troposphere VCD profile of NO2 and O3? The interpolation of pressure in troposphere is also important.
(9) Lines 292-294: By releasing the threshold from 0.75 to 0.5, there might be some potential consequences in data quality etc. Some concerns are raised here.
(10) Line 299: Winter data and data in 2017 around noon are missing, but how about Jan 2018 / other years? It will be great if Jan / Dec data can be included in the revised manuscript.
(11) Lines 326-328: Are there any past studies supporting this conclusion?
(12) Line 332: The data are acquired within SZAs range between 60 degree to 80 degree, but as shown in Figure 6, there are huge AMF difference taking place within this SZA range.
(13) Lines 362-367: Can highlight the contribution from traffic in producing NOx / NO2, and include some possible references in other cities.
(14) Section 3.4: How about other vertical height levels? Why choose 20 m as reference for the comparison here?
(15) Figure 12(a): Obviously, these points in the figures are scattered, therefore not much correlation / statistical relationship can be established. Therefore, the "CORR=-0.33" figure here is not meaningful. Please remove this, or replace by other statistical metrics. The conclusion of this manuscript should be modified as well.
(16) Line 383: The conclusion here (peak obtained at around 50 degree) is mainly due to the positioning of the instrument. Therefore, not much conclusion can be derived too.
(17) Figure 17: After obtaining these plots (or correlation charts), some extensions and summary / potential association with meteorological variables should be discussed / explored.
(18) Section 5 (Conclusion): The future goals and vision based on the current study or instrumental set-up are not explicitly laid down. Need some details within the Conclusion paragraph.
Typos:
Caption of Figure 8: during "the" days
Line 342: Also --> Further
Line 346 and 366: "systematically"
Line 408: and --> than
Line 436: Indeed, OMI O3 VCDs in 2017 are also lower than...
Line 453: in non-polluted regions
Line 488: which can provide more
Line 489: and allow us to better study
I am very looking forward to reading the revised manuscript, and in general, the manuscript is well written, with good sentence structure and meaningful contents.
Round 2
Reviewer 1 Report
The authors did a great job in revising the manuscript. With pleasure I recommend acceptance of the revised manuscript for publication.
Author Response
We would like to thank the reviewer for his/her important suggestions. The manuscript has certainly improved.
Reviewer 3 Report
The revised manuscript looks much better, there are just some minor points to update / revise.
(1) For Table 2, a short description should be provided before ref [49] for O4 and H2O respectively.
(2) Thanks for providing the sensitivity test and corresponding scatter plots when the threshold changes from 0.75 to 0.5. Please kindly include a statement in the manuscript that this factor will merely / barely affect the eventual retrieval results.
(3) Figures 10 and 11 - the correlation values should not appear in the discussion / graph, because these values are not meaningful anyway
(4) The current conclusion paragraph seems to be rather long. Please shorten it before publication.
